# Gastrointestinal Parasites of Dogs in Egypt: An Update on the Prevalence in Dakahlia Governorate and a Meta-Analysis for the Published Data from the Country

**DOI:** 10.3390/ani13030496

**Published:** 2023-01-31

**Authors:** Ibrahim Abbas, Hanadi B. Baghdadi, Mohamed Abdo Rizk, El-Sayed El-Alfy, Bassem Elmishmishy, Mayada Gwida

**Affiliations:** 1Parasitology Department, Faculty of Veterinary Medicine, Mansoura University, Mansoura 35516, Egypt; 2Biology Department, College of Science, Imam Abdulrahman Bin Faisal University, Dammam 31113, Saudi Arabia; 3Basic and Applied Scientific Research Center (BASRC), Imam Abdulrahman Bin Faisal University, Dammam 31113, Saudi Arabia; 4Department of Internal Medicine and Infectious Diseases, Faculty of Veterinary Medicine, Mansoura University, Mansoura 35516, Egypt; 5Hygiene and Zoonoses Department, Faculty of Veterinary Medicine, Mansoura University, Mansoura 35516, Egypt

**Keywords:** gastrointestinal parasites, *Toxocara canis*, dogs, zoonoses, meta-analysis, Egypt

## Abstract

**Simple Summary:**

Dogs are hosts for several gastrointestinal (GIT) parasites that pose potential threats for health of humans and animals. Herein, various GIT parasites in feces of stray dogs in Dakahlia governorate, Egypt, were surveyed. Interestingly, the prevalence greatly declined compared to what had been detected in the latest surveys from Dakahlia published 40 years ago; nonetheless, a few parasites, including *Toxocara canis*, remain prevalent. Various meta-analyses were also conducted to combine our findings with findings of earlier surveys on dogs from Egypt, and the results highlighted the need for a close collaboration between veterinary and public health authorities in Egypt in a “One Health” approach.

**Abstract:**

Since the last survey on gastrointestinal (GIT) parasites infecting dogs in Dakahlia governorate, Egypt, was published 40 years ago, the present study detected various GIT parasites in feces of 78 stray dogs in this governorate. Twenty-one dogs (35.9%) had eggs/oocysts of eight different parasites including *Toxocara canis* (19.2%), *Toxascaris leonina* (2.6%), hookworms (1.3%), *Taenia* species (5.1%), *Dipylidium caninum* (2.6%), *Cystoisospora canis* (5.1%), *Cystoisospora ohioensis* (2.6%), and *Neospora caninum*-like oocysts (1.3%). These results were combined in various meta-analyses with findings of all published surveys on GIT parasites of dogs in Egypt to underline the potential parasitic zoonoses from dogs in the country. Feces and/or gastrointestinal tracts of 19,807 dogs from various Egyptian governorates, but particularly Cairo, have been microscopically tested in 182 datasets published between 1938 and 2022, revealed during our systematic database search. *Toxocara canis*, interestingly, displayed a twofold higher pooled prevalence (24.7%) when compared to the published global pooled prevalence for *T. canis*, indicating that dogs represent a major risk for toxocariasis in humans from Egypt. *Dipylidium caninum* (25.4%) as well as various *Taenia* species (17.1%) also displayed high pooled prevalences. On the contrary, lower pooled prevalence was estimated for the most important zoonotic taeniid “*Echinococcus granulosus*” (2.4%) as well as for hookworms (1.8%) in comparison to what has been published from other countries in the region. Relatively high prevalences were estimated for three protozoa detected in dogs and are common to infect children in Egypt; *Cryptosporidium* (5.5%), *Giardia* (7.4%), and *Entamoeba histolytica* (9.8%). In general, the pooled prevalence estimated for various parasites detected in dogs from Egypt has decreased in the recent years, sometimes by as much as one-fifth, but this great decline is statistically insignificant, which should alert the veterinary and public health authorities to continue their efforts for controlling these parasites in a “One Health” approach.

## 1. Introduction

Around 15 million dogs live as strays in Egypt and represent the majority of the dog population in this country (https://learningenglish.voanews.com/a/egypt-s-street-dogs-getting-chance-at-better-life/5297129.html, accessed on 15 October 2022). Those dogs can roam everywhere in rural and urban areas as well, spread various gastrointestinal parasites with potential zoonoses in the environment [1]. Many of these parasites are very serious, e.g., *Echinococcus*, which circulate in various dog–human and dog–animal cycles. Echinococcosis affects more than a million people worldwide at any one time, and it results in around 19,300 fatalities per year [2]. Although being underreported, echinococcosis has increasing prevalence patterns among Egyptians in the recent years, reviewed in [3]. For example, 45 cases were reported among patients who visited Tanta hospital in Gharbia governorate between 2012 and 2014; many of them had multiple cysts in their liver and/or lungs. Another 46 cases were documented in 2016–2018 in a hospital in Cairo. More recently, 10 CE cases were detected in children who visited a hospital in Cairo [3]. In addition, some surveys on animals are alarming with variable prevalence rates reaching up to 50.0% in camels and 27.0% in sheep, reviewed in [3].

Several surveys have been published describing parasitic infections of dogs in various Egyptian governorates. The majority of these surveys have recruited a limited number of dogs, and findings of these surveys are mostly fragmentary. This highlights the need for a meta-analysis study combining results of these surveys to provide a comprehensive overview on various parasites infecting dogs and their potential zoonoses in Egypt. Moreover, surveys from certain governorates are old and need updating. For example, the last survey on dogs from Dakahlia (the largest governorate in the Nile Delta) was published 40 years ago [4].

The objective of the present study was to provide the first meta-analysis investigation on various gastrointestinal parasites infecting dogs in Egypt, and to update the prevalence of GIT parasites infecting dogs in Dakahlia governorate. Results would be useful in highlighting the most important parasitic zoonoses from dogs in Egypt.

## 2. Materials and Methods

### 2.1. Experimental Study

Fresh fecal samples were collected over a period of 6 months from 78 dogs of different ages and sexes that lived as strays in Mansoura city suburbs, Dakahlia governorate, Egypt. Around 10 g of fresh feces were collected from each dog. Samples were grossly examined, then processed using the standard sedimentation test in combination with the modified Wisconsin sucrose flotation test [5]. Samples that had coccidian oocysts were kept in potassium dichromate 2.5% at room temperature for oocyst sporulation. *Cryptosporidium* oocysts were tested using the modified Ziehl-Neelsen’s staining technique [6]. Morphometrics of the revealed parasitic stages were detected using a binocular microscope (Carl Zeiss, Oberkochen, Germany) equipped with a calibrated ocular micrometer as well as a 3 megapixel camera (Amscope^®^, Irvine, CA, USA). The identity of the revealed coccidian oocysts was detected according to [7].

### 2.2. Meta-Analysis

#### 2.2.1. Reports Collection, Testing for Eligibility, and Data Extraction

Various databases (PubMed, Scopus, ScienceDirect, and Google Scholar) were searched by the two authors (IA, MG) for studies listing the GIT parasites of dogs in Egypt. Several keywords were used in various combinations and linked using the Boolean operators “AND” and “OR”. The keywords included gastrointestinal parasites, helminths, *Toxocara canis, Toxascaris leonina,* hookworms*, *taeniids*, Dipylidium caninum,* protozoa, *Isospora*, *Neospora caninum, Cryptosporidium*, dogs, and Egypt. Websites of the scientific networks “ResearchGate and Academia” were also included in our search. The website of the Egyptian knowledge bank (http://www.ekb.eg, accessed on 30 September 2022) was searched to collect papers from Egypt published in local journals, which were not available in electronic copies. The collected articles were screened for inclusion by EE and BE and articles with disagreement were discussed with IA and MG. Only papers that had been published as research articles were considered. Articles were also defined as eligible when the study was conducted in Egypt, found positive samples for any of the GIT parasites in dogs, and had a defined number of tested as well as positive samples. Articles that did not meet these criteria were considered ineligible, e.g., articles listing non-GIT parasites in dogs, articles of non-original contributions (e.g., reviews), and articles with inappropriate methodologies. Data were extracted from the eligible studies and organized in Microsoft Excel^®^ (Redmond, WA, USA) spreadsheets (version 2020) by EE and BE and any disagreement was resolved by consensus. The following information was extracted: study region/governorate, dogs’ life style, sample size, number of positives, parasites detected, and the detection method. A few authors of articles with unclear data were contacted.

#### 2.2.2. Data Analysis

The extracted data were used for the meta-analysis conducted using the software Open Meta[Analyst] [8], and all analyses were established based on a 95% confidence interval (95% CI). Because of the high heterogeneity (estimated based on the *I^2^* statistic) among the included datasets, pooled estimates representing the prevalence of various GIT parasites were computed using the random effects model based on the DerSimonian–Laird method. The heterogeneity among studies was considered high when the *I^2^* value exceeded 50%. Various subgroup analyses were conducted to investigate the prevalence variation according to the dogs’ life style (strays or pets) and the regional origin of dogs sampled. The Egyptian governorates were assigned into 4 main regions (Nile Delta, Middle Egypt, Coastal governorates, and Southern governorates). Since the year of samples’ collection has not been provided in most reports from Egypt, datasets were classified into 2 groups according to the publishing year (before and after the year 2010) to detect the prevalence variation over time. Publication bias was not estimated for the collected data because it is not considered relevant for prevalence studies [9]. Data were also statistically tested using the software GraphPad Prism (version 6), and various tests were employed. For example, the one-way ANOVA was used in combination with the Kruskal–Wallis test to test the significance of variation among various regions. The unpaired *t* test coupled to the Mann–Whitney test were used to test the significance of variation according to the dogs’ life style. Variations were considered significant when the *p*-value ≤ 0.05.

## 3. Results

### 3.1. Experimental Study

Eggs/oocysts of eight different GIT parasites were identified in feces of 78 stray dogs from Dakahlia governorate, giving rise to an overall prevalence of 35.9% (28/78). *Toxocara canis* was the most frequently detected parasite; *T. canis* eggs were detected in 15 (19.2%) samples. Other helminths eggs were also observed, including *T. leonina* (2.6%), hookworms (1.3%), taeniid eggs (5.1%), and *D. caninum* (2.6%) (Table 1).

**Table 1 animals-13-00496-t001:** Prevalence of different parasites in feces of 78 stray dogs from Dakahlia governorate, Egypt.

Identified Parasites	No. Positive	%	Single Infection	Mixed Infection
*Toxocara canis*	15	19.2	13	2
*Toxoscaris leonina*	2	2.6	2	--
Hookworms	1	1.3	1	--
Taeniid eggs	4	5.1	3	1
*Dipylidium caninum*	2	2.6	2	--
*Cystoisospora canis*	4	5.1	1	3
*Cystoisospora ohioensis*	2	2.6	1	1
*Neospora caninum-*like oocysts	1	1.3	1	--

Two species of the coccidium *Cystoisospora* were detected. *Cystoisospora canis* oocysts were found in four (5.1%) samples. Oocysts (*n* = 30) were ovoid and measured 37.5–44.2 × 32.8–36.3 µm. The oocyst wall was smooth, pale tan, and 1.5 µm thick. No micropyle or oocyst residuum were detected. Sporulated oocysts had two ellipsoidal to ovoid sporocysts with no stieda body (Appendix A). Oocysts of the second type, *Cystoisospora ohioensis* [8], were found in two (2.6%) samples. Oocysts (*n* = 20) were broadly ovoid to subspherical with smooth colorless walls (0.8µm thick) and measured 20–25.6 × 18.5–24.1 µm. No micropyle or polar granules were noticed. The oocyst residual body was absent. Sporocysts were ellipsoidal to ovoid and devoid of the stieda body (Appendix A).

*Neospora caninum*-like oocysts were detected in fecal floats of a single dog; a few oocysts (5–10) were noticed in each microscopic field. Oocysts (*n* = 30) were spherical to subspherical, measured 10.2–11.5 µm and had colorless walls (Appendix A). Sporulated oocysts had two sporocysts each with four sporozoites. No polar granules, stieda, or oocyst residual bodies were detected (Appendix A). In addition, no *Cryptosporidium* oocysts were detected in any stained fecal smear. Although not common, mixed infections were observed in three samples in the form of dual *T. canis-D. caninum*, *T. canis-C. canis*, or *Taeniid-C. canis* infection.

### 3.2. Meta-Analysis

Thirty-six studies in all (including eight in journals not available as electronic copies) were defined as eligible for inclusion in the meta-analysis (Table 2; Figure 1) and included 182 datasets describing prevalence of seven helminths (139 datasets) and five protozoa (43 datasets) inhabiting the GIT of dogs in Egypt. In total, 19,807 dogs were sampled in surveys mostly from Cairo (the Capital, 14,706 dogs), but also some other governorates including Dakahlia (1117 dogs). The number of samples collected from housed dogs (9679) was not far from that of stray dogs (10,076). The included studies tested the sampled dogs employing either fecal examination to detect eggs/oocysts (10,487 dogs) or examination of the intestine of the necropsied dogs (8635 dog) to retrieve the adult worms. PCR has not been used in any of the included studies to diagnose the GIT parasite found in the tested samples. Values of the overall and regional pooled prevalences for the included parasites are summarized through Table 3, Table 4, Table 5, Table 6, Table 7, Table 8, Table 9 and Table 10.

**Table 2 animals-13-00496-t002:** Characteristics of studies included in various meta-analyses conducted in the present study. Abbreviation: FE, fecal examination; NS, not stated.

Governorate	Publication Year	Source of Samples Mode of Life	Identification Stage, Method	No. Tested	Identified Parasites (Prevalence)	Reference
Cairo	1938	Stray	Adult, Necropsy	150	*E. granulosus* (10%), *Heterophyes heterophyes* (59%)	[10]
Alexandria				100	*E. granulosus* (2%), *H. heterophyes* (75%)	
Upper Egypt				70	*E. granulosus* (3%)	
Cairo	1974	Stray	Adult, Necropsy	570	*E. granulosus* (3.9%)	[11]
Alexandria	1979	Stray	Eggs, FE	459	*T. canis* (23.7%)	[12]
		Pet		128	*T. canis* (13.3%)	
Dakahlia	1980	Stray	Adult, Necropsy	58	*T. canis* (84.7%), *Taenia hydatigena* (28.2%), *D. caninum* (62.4%), *E. granulosus* (1.2%), *E. perfoliatus* (4.7%), *H. heterophyes* (12.9%), *Haplorchis yokogawai* (10.6%), *P. vivax* (9.4%)	[4]
Dakahlia	1982	Stray	Oocysts, FE	125	*Isospora canis* (97.6%), *Isospora ohioensis* (39.2%)	[13]
Dakahlia	1982	Stray	Adult, Necropsy	289	*E. granulosus* (1.4%)	[14]
Cairo	1988	Stray	Oocysts, FE	110	*Eimeria canis* (1.8%), *I. canis* (6.4%), *I.* *ohioensis (2.7%)*, *Hammondia heydorni* (2.7%), *Sarcocystis* spp. (4.5%)	[15]
Ismailia	1991	Stray	Adult, Necropsy	20	*T. canis* (10%), *T. leonina* (30%), *D. caninum* (60%)	[16]
Cairo	1991	Stray	Adult, Necropsy	5000	*Taenia pisiformis*	[17]
Menoufiya	1993	Stray	Adult, Necropsy	165	*T. canis* (72.7%)	[18]
Ismailia	1995	Stray	Eggs/oocysts, FE	685	*T. canis* (25.5%), *T. leonina* (32.1%), hookworm (1.3%), *Taenia* spp. (10.2%), *D. caninum* (17.5%), *Spirometra* spp. (0.2%), *H. heterophyes* (8.5%), *I. canis* (4.6%), *I. ohioensis* (13%), *Isospora spp.* (17.6%), *Sarcocystis* spp. (2.3%), *Cryptosporidium* spp. (3.8%), *Giardia* spp. (8.3%)	[19]
Beni Suef	1998	Stray	Adult, Necropsy	87	*T. canis* (18.4%), *T. leonina* (57.5%), *T. pisiformis* (19.5%), *T. hydatigena* (17.2%), *D. caninum* (43.7%), *E. perfoliatus* (3.45%), *Spirocerca lupi* (12.6%)*, Mesostephanus appendiculatus* (3.45%), *Mesostephanus melvi* (4.6%), *Mesostephanus fajardensis* (4.6%), *P. vivax* (4.6%)	[20]
Aswan	1999	Stray dogs	Oocysts, FE	27	*Cryptosporidium* spp. (11.1%)	[21]
Sharkia	2002	Stray	Adult, Necropsy	29	*T. canis* (51.7%), *S. lupi* (3.4%), *Rictularia affinis* (6.9%), *T. pisiformis* (24.1%), *D. caninum* (86.2%), *Spirometra erinacei* (3.4%), *Diplopylidium nolleri* (3.4%), *H. heterophyes* (10.3%), *P. vivax* (6.9%)	[22]
Assiut	2003	Stray	Adult, Necropsy	70	*T. pisiformis* (18.5%), *T. hydatigena* (15.7%), *Taenia ovis* (4.2%), *Taenia taeniaeformis* (1.4%), *E. granulosus* (1.4%)*, D. caninum* (45.7%),	[23]
Dakahlia	2007	Stray	Adult, Necropsy	540	*E. garnulosus* (5%)	[24]
Cairo	2007	Stray	Adult, Necropsy	50	*T. canis*, *D. caninum*, *E. granulosus* (16%)	[25]
Cairo	2007	Stray	Adult, Necropsy	166	*T. canis* (39.2%), *T. leonina* (30%), hookworms (25.9%), *Taenia* spp. (36.1%), *D. caninum* (32.5%), *E. granulosus* (1.2%), *Cystoisospora* spp. (15.7%), *E. histolytica* (34.9%), *Sarcocystis* spp. (1.8%)	[26]
		Housed	Eggs/oocysts, FE	60	*T. canis* (16.7%), *T. leonina* (30%), hookworms (6.7%) *Taenia* spp. (13.3%), *D. caninum* (5%), *Cystoisospora* spp. (10%), *E. histolytica* (21.7%), *Sarcocystis* spp. (1.7%)	
Cairo	2008	Housed	Eggs, FE	500	*T. canis* (6%), *D. caninum* (40%), *E. granulosus* (1.8%)	[27]
Beni Suef	2008	Stray dogs	Eggs, FE	200	*T. canis* (10%), *A. caninum* (5%), *Taenia* spp. (3%), *D. caninum* (3%), *H. heterophyes* (1.5%)	[28]
NS	2009	Pet	Eggs/oocysts, FE	3000	*T. canis* (9.8%), *E. granulosus*, *D. caninum*, *H. heterophyes*	[29]
Giza	2009	Stray dogs	Egg/oocysts, FE	27	*T. canis* (37%), *T. leonina* (33.3%), *A. caninum* (18.5%), *Taenia* spp. (33.3%), *D. caninum* (37%), *Isospora* spp. (14.8%), *Sarcocystis* spp. (14.8%), *Cryptosporidium* spp. (18.5%), *Giardia* spp. (14.8%)	[30]
Kalubiya	2010	Stray dogs with diarrhea	Oocysts, FE	20	*Cryptosporidium* spp. (50%), including 2 that were molecularly identified as *C. parvum*	[31]
Giza	2010	Stray	Adult, Necropsy	25^@^	*T. canis* (56%), *T. leonina* (8%), *Ascaris lumbricoides* (8%)	[32]
NS	2011	Stray	^#^ Eggs, FE	53	*T. canis* (35.8%)	[33]
		Housed		45	*T. canis* (21.3%)	
Alexandria	2012	Stray	Eggs/oocysts, FE	33	*T. canis* (15.1%), *A. caninum* (21.2%), *Echinococcus* spp. (24.2%), *E. perfoliates* (9.1%), *D. caninum* (60.6%), *H. heterophyes* (27.2%), *Ascocotyle* spp. (3.03%)	[34]
Alexandria	2014	Housed	Eggs/oocysts, FE	120	*T. canis* (0.8%), *A. caninum* (1.7%), *T. vulpis* (0.8%), *Cystoisospora canis* (4.2%), *Giardia* spp. (1.7%)	[35]
		Police		60	*T. canis* (5%), *T. leonina* (1.7%), *C. canis* (3.3%), *Giardia* spp. (31.7%), *Entamoeba histolytica* (18.3%), *Cryptosporidium* spp. (1.7%)	
Cairo	2014	Stray	Eggs/oocysts, FE	180	*T. canis* (46.1%), *T. leonina* (3.9%), *T. vulpis* (34.4%), *Capillaria* spp. (12.2%), *Taenia* spp. (10%), *D. caninum* (22.8%), *Isospora* spp. (37.8%)	[36]
Ismailia	2015	Stray	Adult, Necropsy	50	*T. canis* (20%), *T. leonina* (10%), *S. lupi* (10%), *R. affinis* (8%), *T. hydatigena* (10%), *D. caninum* (100%), *P. vivax* (4%), *M. appendiculatus* (16%), *M. melvi* (6%), *Mesostephanus* spp. (2%), *E. liliputans* (16%), *H. dispar* (14%), *P. genata* (20%), *Pygidiopsis summa* (6%), *Ascocotyle rara* (4%), *Phagicola longus* (6%), *Phagicola longicollis* (4%), *Metagonimus yokogawai* (4%), *Haplorchis pumilio* (6%), *Apophallus donicus* (4%)	[37]
Kalubiya	2015	Military (30), housed (60), stray (30)	Eggs/oocysts, FE	130	*T. canis* (5.4%), *T. leonina* (3.1%), *Ancylostoma* spp. (6.2%), *Taenia* spp. (2.3%), *D. caninum* (1.5%), *Heterophyes* spp. (3.9%), *Paragonimus* spp. (0.8%), *Cryptosporidium* spp. (5.4%), *Blastocystis* spp. (3.1%), *Entamoeba canis* (0.8%), *Cyclosppora caytanensis* (0.8%)	[38]
Cairo	2015	Housed	Eggs, FE	3864	*T. canis* (3%)	[39]
Cairo, Giza	2016	Housed	Eggs/oocysts, FE	395	*T. canis* (0.3%), *T. leonina* (5.8%), *T. vulpis* (3.3%), *Cryptosporidium* spp. (10.1%), *Giardia* spp. (0.5%), *Entamoeba histolytica/Entamoeba dispar* (5.6%)	[40]
Sohag, Luxor	2018	Housed	Eggs, FE	120	*T. canis* (23.3%), *T. leonina* (4.2%), *A. caninum* (3.3%), *D. caninum* (1.7%)	[41]
Sharkia	2018	Housed	Oocysts, FE	50	*Cryptosporidium* spp. (34%)	[42]
			Nested-PCR		*Cryptosporidium* spp. (24%)	
Alexandrina, Ismailia, Menoufiya, Sohag	2019	Pet	Eggs, FE	296	*T. canis* (53.1%)	[43]
Various	2021	Pet	Eggs/oocysts, FE	986	*Giardia* spp. (8.5%)	[44]
Dakahlia		Stray	Eggs/oocysts, FE	78	*T. canis* (19.2%), *T. leonina* (2.6%), hookworms (1.3%), *Taenia* spp. (5.1%), *D. caninum* (2.6%), *C. canis* (5.1%), *C. ohioensis* (2.6%), *N. caninum-like oocyst* (1.3%)	Present study

^#^ *Toxocara canis* eggs have also been noticed in hair samples from 26.6% of 64 stray and 10.7% of 54 domestic dogs.

## 4. Discussion

In the present study, feces of 78 stray dogs from Dakahlia governorate, Egypt, were screened for various GIT parasites, and some parasites that are widely known as potential agents of zoonosis (e.g., *T. canis*, hookworms, and *D. caninum*) were detected in variable infection rates. It is worthy to mention that the prevalence of these parasites greatly declined from what was reported 40 years ago from stray dogs in Dakahlia [4]. Our findings from dogs in Dakahlia were combined with findings of the earlier surveys on dogs from Egypt in various meta-analyses to illustrate the role of dogs in the epidemiology of dog–man transmitted parasites throughout Egypt. The following sections cover the most common parasitic zoonoses from dogs in Egypt based on findings of various meta-analyses conducted in the present study.

### 4.1. Common Nematodes in Dogs from Egypt

*Toxocara canis* is one of the major zoonotic helminths worldwide. Eggs of this nematode were detected in feces of 19.2% of the 78 screened dogs from Dakahlia. In total, 11,477 dogs from various Egyptian governorates were sampled and 1406 were found infected, giving rise to a pooled prevalence (24.7%, 95% CI: 21.1–28.3%) that is more than double the estimated *T. canis* global prevalence (11.1%) [45], suggesting a higher risk for human health in Egypt (Table 3; Figure 2). The pooled prevalence did not significantly vary over time (*p*-value = 0.7390); however, the estimated pooled prevalence for the published datasets after the year 2010 was lower (21.4%, 17.5–25.4%) than that for datasets published before 2010 (27.0%, 19.8–34.3%). In general, dogs in Africa and the Middle East, where Egypt is located, have higher *T. canis* prevalence than any other region worldwide due to socio-economic, environmental, and climatic factors [45]. While dogs from the Nile Delta had the highest *T. canis* prevalence (42.6%, 8.2–77.1%) in Egypt when compared to the other three regions (Table 3), prevalence variations between regions were insignificant (*p*-value = 0.231), assuming that dogs have similar susceptibility to *T. canis* infection throughout Egypt. This was also true for prevalence variation between stray and housed dogs: no significant difference (*p*-value = 0.141) was found; however, stray dogs (30.3%) had a higher prevalence than housed dogs (16.4%) (Table 3). In Egypt, many dogs spend a time in their life as strays before being housed. A recent report by Abdel Aziz et al. [43] underlines this assumption. The authors examined 296 pet dogs admitted to veterinary hospitals in four different Egyptian governorates and found a very high *T. canis* prevalence (53.1%). This report is alarming since keeping a dog has become a popular trend among youth particularly in urban cities in Egypt. On the other hand, the high prevalence in stray dogs reflects a significant level of environmental contamination with *T. canis* eggs, which have been detected in high prevalences in soil samples from various Egyptian governorates [46]. Humans can be accidentally infected through ingesting *T. canis* eggs, and infected humans often develop ocular toxocariasis or visceral larval migrans with subsequent allergic and neurological disorders [47]. Although a high prevalence of *T. canis* antibodies has been observed in several surveys from humans in Egypt, toxocariasis is underestimated [48]. Nevertheless, *T. canis* represents a significant zoonosis in this country taking into consideration the high prevalence of *T. canis* in dogs from Egypt.

**Table 3 animals-13-00496-t003:** Overall and regional pooled prevalence of *Toxocara canis* in dogs from various Egyptian regions, and variabilities according to dog life style and detection method.

Parameter	No. Data Sets	No. Tested	No. Positive	Pooled Estimate % Based on 95% CI	Heterogeneity *I*^2^%
Overall prevalence	29	11,477	1406	24.7 (21.1–28.3)	98.67
Prevalence variation over time					
Before 2010	13	6151	976	27.0 (19.8–34.3)	98.53
After 2010	14	5326	430	21.4 (17.5–25.4)	98.00
Regional prevalence					
Nile Delta	6	925	248	42.6 (8.2–77.1)	99.33
Dakahlia	3	703	106	35.4 (−12.9–83.7)	99.48
Menoufiya	2	193	127	49.9 (2.7–96.2)	96.53
Sharkia	1	29	15	51.7 (33.5–69.9)	NA
Middle Egypt	10	622	8322	13.6 (9.9–17.2)	98.32
Cairo	7	8165	605	14.3 (10.2–18.4)	98.84
Giza	1	27	10	37.0 (18.8–55.3)	NA
Kalubiya	2	130	7	6.4 (−3.1–16.0)	61.20
Coastal governorates	9	1762	427	23.1 (12.4–33.7)	97.34
Alexandria	5	962	218	21.0 (5.5–36.5)	98.10
Ismailia	4	800	209	25.6 (14.2–36.9)	81.39
Southern governorates	4	109	468	30.9 (8.1–53.7)	97.62
Beni-Suef	2	287	33	11.4 (7.0–15.9)	21.78
Luxor	1	120	30	25.0 (17.3–32.7)	NA
Sohag	1	61	46	75.4 (64.6–86.2)	NA
Dog life style					
Stray	16	2834	737	30.3 (19.3–41.2)	98.44
Housed	13	8643	669	16.4 (12.8–19.9)	98.30
Detection method					
Necropsy	9	1639	346	33.3 (18.3–48.3)	99.02
Fecal examination	20	9835	1060	20.7 (17.1–24.2)	98.37

NA, not applicable.

The other Ascarid “*T. leonina*” common to infect dogs and cats worldwide has been examined in 21 datasets from dogs in Egypt (Table 4). When compared to *T. canis*, the parasite displayed a much lower pooled prevalence (2.8%, 2.0–3.6%) in 7429 tested dogs in Egypt, and the prevalence did not significantly (*p*-value = 0.5772) vary over time (Table 4; Appendix A). This very low prevalence is consistent with that has been estimated for 119,317 dogs worldwide (2.9%) [49], and can be attributed to the limited routes for *T. leonina* transmission among dogs in comparison to *T. canis* [49]. *Toxascaris leonina* prevalence did not significantly differ according to the Egyptian region studied (*p*-value = 0.313) or the life style of dogs (*p*-value = 0.645). Although not common, there have been a few cases of *T. leonina* infections in humans worldwide [50].

**Table 4 animals-13-00496-t004:** Overall and regional pooled prevalence of *Toxascaris leonina* in dogs from various Egyptian regions, and variabilities according to dog life style and detection method.

Parameter	No. Data Sets	No. Tested	No. Positive	Pooled Estimate % Based on 95% CI	Heterogeneity *I*^2^%
Overall prevalence	21	7429	332	2.8 (2.0–3.6)	96.13
Prevalence variation over time					
Before 2010	11	2399	285	5.2 (3.5–7.0)	97.83
After 2010	10	5030	47	2.5 (1.0–4.0)	82.00
Regional Prevalence					
Nile Delta	4	732	2	0.1 (0.1–0.4)	0.00
Dakahlia	3	703	2	0.2 (−0.3–0.7)	10.15
Sharkia	1	29	0	1.7 (−2.9–6.2)	NA
Middle governorates	9	5322	43	0.8 (0.2–1.4)	84.19
Cairo	6	5165	30	0.6 (0.0–1.1)	84.77
Kalubiya	2	130	4	3.9 (−4.3–12.1)	61.74
Giza	1	27	9	33.3 (15.6–51.1)	NA
Coastal governorates	5	968	232	14.0 (0.9–28.9)	98.65
Alexandria	2	213	1	0.6 (0.4–1.7)	0.00
Ismailia	3	755	231	23.6 (6.7–40.6)	91.34
Southern governorates	3	407	55	19.0 (3.3–34.7)	98.33
Beni-Suef	2	287	50	28.6 (−27.5–84.7)	99.14
Luxor	1	120	5	4.2 (0.6–7.7)	NA
Dog life style					
Stray	13	2180	299	7.8 (5.2–10.4)	97.46
Housed	8	5249	33	0.7 (0.1–1.2)	80.32
Detection method					
Necropsy	14	6118	271	2.2 (0.6–3.8)	95.45
Fecal examination	7	1311	61	4.3 (2.6–6.1)	96.61

NA, not applicable.

In the present study, a single dog out of 78 tested from Dakahlia had hookworm eggs in his feces. In total, 20 datasets from Egypt that tested 6923 dogs for hookworms were revealed during our search; 139 dogs were found infected, resulting in a pooled prevalence of 1.8% (1.0–2.6%) (Table 5; Appendix A). This prevalence is much lower when compared to that detected for dogs in some African countries [51]. A very high pooled prevalence (41.0%) has also been estimated for dogs in Asia [52]. Although being much declined, the pooled prevalence did not significantly (*p*-value = 0.8497) vary when datasets published after the year 2010 (0.9%) on dogs from Egypt were compared to those published earlier (3.9%). In Egypt, comparable prevalences were detected for dogs from various regions with no significant differences (Table 5). Likewise, the life style of dogs had no significant effect (*p*-value = 0.967) on the prevalence (Table 5). Various hookworm species can infect dogs worldwide, e.g., *Ancylostoma caninum*, *Ancylostoma braziliense*, *Ancylostoma ceylanicum*, and *Uncinaria stenocephala*. These species can cause anemia and hypoproteinemia with varying degrees in dogs [53]. Based on morphometrics of eggs recovered from the tested fecal samples, *A. caninum* has been identified in the majority of surveys on dogs from Egypt. All canine hookworms are zoonotic [53], and hookworm infections were detected in patients from Egypt early in the 1990s [54]. Additionally, hookworms have been identified as being responsible for acute and recurrent abdominal pain in 11 of 95 patients from Egypt [55] Hookworm eggs and/or larvae have also been detected in soil samples as well as wastewater in Egypt [56].

**Table 5 animals-13-00496-t005:** Overall and regional pooled prevalence of hookworms in dogs from various Egyptian regions, and variabilities according to dog life style.

Parameter	No. Data Sets	No. Tested	No. Positive	Pooled Estimate % Based on 95% CI	Heterogeneity *I*^2^%
Overall prevalence	20	6923	139	1.8 (1.0–2.6)	82.84
Prevalence variation over time					
Before 2010	10	1893	82	3.9 (1.8–5.9)	86.63
After 2010	10	5030	57	0.9 (0.2–1.6)	70.91
Regional prevalence					
Nile Delta	4	732	12	1.5 (0.6–2.4)	0.0
Dakahlia	3	703	11	1.5 (0.6–2.4)	3.40
Sharkia	1	29	0	1.7 (−2.9–6.2)	NA
Middle governorates	8	4822	95	2.2 (0.9–3.5)	91.63
Cairo	5	4665	82	1.6 (0.3–3.0)	94.34
Giza	1	27	5	18.5 (3.9–33.2)	NA
Kalubiya	2	130	8	6.8 (−1.6–15.2)	51.37
Coastal governorates	5	968	18	1.5 (0.0–2.9)	50.29
Alexandria	2	213	9	9.9 (−9.6–29.5)	87.31
Ismailia	3	755	9	1.3 (0.5–2.1)	0.0
Southern governorates	3	401	14	2.7 (0.1–5.6)	71.67
Beni-Suef	2	281	10	2.6 (−1.7–6.9)	83.84
Luxor	1	120	4	3.3 (0.1–6.5)	NA
Dog life style					
Stray	14	2204	90	2.9 (1.4–4.4)	75.97
Housed	6	4719	49	1.0 (0.2–1.8)	91.92

NA, not applicable.

### 4.2. Common Cestodes in Dogs from Egypt

Various cestodes can circulate in dog–human/animal cycles. Dogs are definitive hosts for several taeniids, and infections can be diagnosed through detection of the gravid segments and/or eggs in dog feces; however, eggs are not discriminative to the species level [57]. In Egypt, 12,021 dogs were tested for *Taenia* species in 19 datasets, and 3926 were found positive, yielding a pooled prevalence of 17.1% (7.2–27.7%) (Table 6; Appendix A). Datasets published before 2010 displayed a five times higher pooled prevalence (26.4%) than those published after 2010 (5.2%), with a significant variation (*p*-value = 0.0085). The prevalence was approximately four times higher (27.0%) in 6501 dogs tested via intestinal necropsy than in 5520 dogs tested via fecal examination (7.2%), with the difference being statistically insignificant (*p*-value = 0.139). Similarly, insignificant variations were detected in *Taenia* prevalence according to the life style of tested dogs (*p*-value = 0.348) or the Egyptian region studied (*p*-value = 0.565) (Table 6). It is worthy to mention that *T. hydatigena* cysticerci have been frequently observed in small ruminants from Egypt [58]. While no *Taenia multiceps* has been detected in dogs from Egypt [59], *T. multiceps* coenurosis has been reported in an Egyptian woman (40 years old) from Tanta city [60], and the disease is quite common in sheep from Egypt [61].

**Table 6 animals-13-00496-t006:** Overall and regional pooled prevalence of *Taenia* spp. detected in dogs from various Egyptian regions, and variabilities according to dog life style.

Parameter	No. Data Sets	No. Tested	No. Positive	Pooled Estimate % Based on 95% CI	Heterogeneity *I*^2^%
Overall prevalence	19	12,021	3926	17.5 (7.2–27.7)	99.82
Prevalence variation over time					
Before 2010	11	7390	3818	26.4 (5.2–47.4)	99.87
After 2010	8	4631	108	5.3 (2.1–8.6)	87.98
Regional prevalence					
Nile Delta	4	732	40	12.8 (2.5–23.1)	92.83
Dakahlia	3	703	33	10.2 (0.5–20.9)	94.04
Sharkia	1	29	7	24.1 (8.6–39.7)	NA
Middle Egypt	8	9868	3708	20.5 (6.6–47.7)	99.93
Cairo	5	9711	3696	25.2 (12.5–63.0)	99.96
Giza	1	27	9	33.3 (15.6–51.1)	NA
Kalubiya	2	130	3	2.2 (−0.3–4.7)	0.0
Coastal governorates	3	768	57	7.0 (0.5–13.6)	85.56
Alexandria	1	33	0	1.5 (−2.6–5.5)	NA
Ismailia	2	735	75	10.2 (8.0–12.4)	0.96
Southern governorates	3	357	66	26.2 (1.8–54.2)	97.35
Assiut	1	70	28	40.0 (28.5–51.5)	NA
Beni-Suef	2	287	38	19.5 (−13.6–52.6)	97.53
Various *	1	296	37	12.5 (8.7–16.3)	NA
Life style					
Stray	15	7701	3838	20.4 (3.4–37.4)	99.81
Housed	4	4320	88	6.3 (1.0–11.6)	93.06
Detection method					
Necropsy	9	6501	3720	27.0 (0.4–54.5)	99.89
Fecal examination	10	5520	196	7.2 (3.7–10.6)	93.32

* Alexandria, Ismailia, Menoufia, and Sohag; NA, not applicable.

In the present study, 4 out of 78 dogs tested in Dakahlia had taeniid eggs. In addition to the genus *Taenia*, the family Taeniidae also includes the genus *Echinococcus* (canine dwarf tapeworms), and both of these genera have quite similar eggs that cannot be distinguished morphologically [57]. As a result, the possibility of *Echinococcus* infection in the sampled dogs from Dakahlia cannot be ruled out. Sixteen datasets from Egypt have examined the necropsied intestine of 2809 dogs, mostly strays (n = 2309), for *Echinococcus*, and *E. granulosus* was found in 96 dogs, giving rise to a pooled prevalence of 2.4% (1.4–3.5%) (Table 7; Figure 3). Only two datasets have been published after the year 2010, and recruited a small number of dogs (n = 83), giving rise to a high pooled prevalence (11.5%), which is five times higher than that estimated for 2726 dogs tested in 14 datasets published before 2010 (2.3%); nonetheless, the prevalence variation is statistically insignificant (*p*-value = 0.8162). At the regional level, higher infection rates have been detected in dogs from North Africa, e.g., Tunisia (18.4%), Libya (27.8%), Morocco (35.3%), and Sudan (51.0%) [62,63,64,65]. Cystic echinococcosis (CE) caused by *E. granulosus* is a common zoonoses from dogs worldwide. In Egypt, many CE-infected cases have been reported in the recent few years [66]. In animals, CE occurrence has increased over time and the recent surveys are alarming from Egypt [67]. *Echinococcus granulosus senso lato* is a complex of 5 species/11 genotypes involving *E. granulosus sensu stricto* (G1-2 genotypes), primarily circulates in a sheep–dog cycle, and contributes approximately to 88% of the global cases of human CE [68]. However, most of human cases in Egypt have been attributed to the G6 (*Echinococcus canadensis*) underlining the dog–camel transmission cycle [69]. Unfortunately, no studies have been conducted to verify *Echinococcus* genotypes in dogs from Egypt, which would aid in understanding the epidemiology of CE in the country.

A recent report that identified *D. caninum* eggs in stools of 4 out of 996 Egyptian children highlights this parasitic zoonoses from dogs in Egypt [70]. In the present study, *D. caninum* had a high estimated pooled prevalence (25.4%, 20.4–30.4%) in 7675 dogs that had been tested in 22 datasets (Table 8; Figure 4), which could be attributed to the wide dispersal of *Ctenocephalides canis*, the vector for *D. caninum*, among populations of stray dogs in Egypt [71]. Although the estimated pooled prevalence for datasets published after 2010 (20.8%) was two-thirds of that for datasets published before 2010 (30.0%), the variation was insignificant (*p*-value = 0.1588). Stray dogs (38.4%) had a significantly (*p*-value = 0.004) higher prevalence than housed dogs (3.3%). Globally, the prevalence is highly variable and correlated with the life style of dogs, level of flea infestation in dogs, environmental sanitation, and geographical areas [72]. There have been a few cases of human dipylidiasis that have been documented worldwide [72].

**Table 7 animals-13-00496-t007:** Overall and regional prevalence of *Echinococcus granulosus* detected in dogs from various Egyptian regions, and variabilities according to dog life style.

Parameter	No. Data Sets	No. Tested	No. Positive	Pooled Estimate % Based on 95% CI	Heterogeneity *I*^2^%
Overall prevalence	16	2809	96	2.4 (1.4–3.5)	67.98
Prevalence variation over time					
Before 2010	14	2726	88	2.3 (1.3–3.4)	65.24
After 2010	2	83	8	11.5 (−11.2–34.1)	98.36
Regional prevalence					
Nile Delta	4	943	28	2.2 (0.5–3.9)	61.72
Dakahlia	3	914	28	2.3 (0.3–4.3)	74.28
Sharkia	1	29	0	1.7 (−2.9–6.2)	NA
Middle governorates	5 (Cairo)	1436	56	4.0 (1.6–6.3)	82.39
Coastal governorates	4	207	10	3.1 (−0.9–7.2)	68.22
Alexandria	2	133	10	11.9 (−9.8–33.6)	88.35
Ismailia	2	70	0	1.2 (−1.3–3.7)	0.0
Southern governorates	3	227	2	0.8 (−0.3–2.0)	0.0
Assiut	1	70	0	0.7 (−1.2–2.6)	NA
Beni-Suef	1	87	0	0.6 (−1.0–2.1)	NA
NS	1	70	2	2.9 (−1.0–6.8)	NA
Life style					
Stray	15	2309	87	2.5 (1.3–3.8)	69.98
Housed	1	500	9	1.8 (0.6–3.0)	NA

NA, not applicable.

**Table 8 animals-13-00496-t008:** Overall and regional prevalence of *Dipylidium caninum* in dogs from various Egyptian regions, and variabilities according to dog life style.

Parameter	No. Data Sets	No. Tested	No. Positive	Pooled Estimate % Based on 95% CI	Heterogeneity *I*^2^%
Overall prevalence	22	7675	651	25.4 (20.4–30.4)	99.64
Prevalence variation over time					
Before 2010	12	2469	379	30.0 (22.7–37.3)	98.23
After 2010	10	5206	272	20.8 (13.2–28.4)	99.93
Regional prevalence					
Nile Delta	4	732	86	36.7 (15.8–57.6)	99.03
Dakahlia	3	703	61	20.4 (3.9–36.9)	98.52
Sharkia	1	29	25	86.2 (73.7–98.8)	NA
Middle governorates	9	5322	210	6.2 (4.0–8.3)	96.01
Cairo	6	5165	198	7.0 (4.4–9.5)	97.27
Giza	1	27	10	37.0 (18.8–55.3)	NA
Kalubiya	2	130	2	1.9 (−3.2–7.1)	44.27
Coastal governorates	5	986	202	47.3 (2.0–92.6)	99.92
Alexandria	2	213	20	29.8 (−29.3–89)	98.01
Ismailia	3	755	182	58.8 (−5.9–1.00)	99.88
Southern governorates	3	357	76	30.5 (2.5–63.5)	98.01
Beni-Suef	2	287	44	32.0 (−16.8–62.9)	98.20
Assiut	1	70	32	45.7 (34.0–57.4)	NA
Various *	1	296	77	26.0 (21.0–31.0)	NA
Dog life style					
Stray	15	2280	471	38.4 (19.9–57.0)	99.72
Housed	7	5295	180	3.3 (1.6–5.0)	96.24

* Alexandria, Ismailia, Menoufiya, Sohag; NA, not applicable.

### 4.3. Common Trematodes in Dogs from Egypt

Due to their fish-eating habits, dogs are susceptible to infection with trematodes of the family Heterophyidae, of which 29 species can infect humans [73]. In the present study, no eggs of any trematode parasite were observed in the tested fecal samples from 78 dogs in Dakahlia. However, various heterophyids have been detected in some earlier surveys on dogs from Egypt. In 12 datasets, 1780 dogs from Egypt were screened for the heterophyids, and 322 were found infected with *H. heterophyes,* resulting in an estimated high pooled prevalence (18.7%, 11.7–25.7%) (Table 9; Appendix A). This high prevalence is consistent with the frequent occurrence and heavy intensity of heterophyid metacercariae in tissues of grey mullet and Nile Tilapia widely consumed in Egypt [74], which represents a potential health threat for Egyptians. The estimated pooled prevalence for datasets published before 2010 (24.4%) was five times higher than that estimated for datasets published after 2010 (4.6%); however, this variation was insignificant (*p*-value = 0.3010). *Heterophyes heterophyes* infections have been also documented in humans from Egypt [75].

**Table 9 animals-13-00496-t009:** Overall and regional prevalence of *Heterophyes heterophyes* detected in dogs from various Egyptian regions, and variabilities according to dog life style.

Parameter	No. Data Sets	No. Tested	No. Positive	Pooled Estimate % Based on 95% CI	Heterogeneity *I*^2^%
Overall prevalence	12	1780	322	18.7 (11.7–25.7)	98.16
Prevalence variation over time					
Before 2010	8	1539	308	24.4 (14.0–34.9)	98.75
After 2010	4	241	14	4.6 (−0.5–9.8)	78.45
Regional prevalence					
Nile Delta	3	192	14	7.3 (−11.7–16.7)	84.92
Dakahlia	2	163	11	6.3 (−5.7–18.3)	90.73
Sharkia	1	29	3	10.3 (−0.7–21.4)	NA
Middle Egypt	4	400	164	24.3 (1.1–47.5)	98.42
Cairo	2	370	158	45.1 (18.8–71.4)	96.39
Kalubiya	2	130	5	3.8 (0.5–7.1)	0.0
Coastal governorates	3	818	142	36.9 (−10.2–84.0)	99.12
Alexandria	2	133	84	51.6 (4.8–98.3)	96.54
Ismailia	1	685	58	8.5 (6.4–10.6)	NA
Southern governorates	2	270	3	1.2 (0.1–2.4)	0.0
Beni-Suef	1	200	3	1.5 (−0.2–3.2)	NA
NS	1	70	0	0.7 (−1.2–2.6)	NA
Life style					
Stray	11	1680	318	20.2 (12.6–27.8)	98.32
Housed	1	100	4	4.0 (0.2–7.8)	NA

NA, not applicable.

### 4.4. Common Protozoa in Dogs from Egypt

Coccidiosis is common among dogs worldwide. Before the discovery of *N. caninum* in 1998, there was uncertainty about the identity of various coccidian protozoans infecting dogs [76]. Three types of coccidian oocysts can be identified in feces of infected dogs: large-sized oocysts (∼40 μm long) belong to *C. canis*, medium-sized oocysts (∼25 μm long) belong to *C. ohioensis,* and small-sized oocysts (∼10–12 μm long) belong either to *N. caninum* or *Hammondia heydroni* [77]. *Sarcocystis* spp. commonly excreted as sporocysts can also be found in dog feces [7]. A few reports on dog coccidiosis in Egypt are available and some reports were published before the year 1998. Therefore, some oocysts have been given invalid names. For example, Abdul-Magied et al. [13] identified small-sized oocysts in 3 out of 125 dogs from Dakahlia as *Isospora bigemina*. El Ghaysh [15], on the other hand, identified small-sized oocysts in feces of 3 out of 110 dogs from Cairo as *H. heydroni* oocysts. The same species name was given for oocysts collected from feces of three dogs after being fed on camel and buffalo meats; however, dogs started shedding oocysts at different times [78]. Therefore, it is possible that some small-sized oocysts that were detected in earlier reports from Egypt belong to *N. caninum*. Oocysts of *N. caninum* have been detected in a few dogs worldwide [7]. In the present study, *N. caninum*-like oocysts were detected in feces of 1 out of 78 dogs tested in Dakahlia. *Neospora caninum* is a significant cause of abortion in cattle [7]. A recent survey detected *N. caninum* antibodies in 35.0% out of 116 tested sera from aborted cows in Egypt [79]. In addition, *N. caninum*-like oocysts were detected in feces of five out of nine puppies experimentally fed on placentas and brains of aborted foeti from nine aborted cows in Egypt [80].

Unlike *N. caninum*, *Cystoisopspora* spp. (formerly known as *Isospora*) has no clinical relevance, but a few dogs showed weakness, loss of appetite, diarrhea, and dehydration [77]. However, *Cystoisospora* spp. was the most frequently detected protozoa in dogs from Egypt, and it was found in 6 out of 78 dogs tested in Dakahlia. The estimated pooled prevalence (Table 10) for *Cystoisospora* spp. in dogs from Egypt was high (21.2%, −14.2–56.7%) when compared to the reported infection rates from other regions worldwide [77]. In addition, two species have been identified in dogs from Egypt; *C. canis* (23.3%, 15.6–62.2%) had a higher prevalence than *C. ohioensis* (12.2%, 3.7–20.8%) (Table 10). Dogs are definitive hosts for several *Sarcocystis* spp. that can utilize various herbivores as intermediate hosts, causing significant economic losses. Some of these species are common among ruminants in Egypt [81], which suggests the high prevalence of *Sarcocystis* oocysts in feces of dogs in Egypt. However, a few datasets (n = 6) from the country have been published in which fecal samples from 1126 dogs were examined and *Sarcocystis* oocysts were detected in 29 with a very low pooled prevalence (2.0%, 0.8–3.2%) (Table 10). Since *Sarocystis hominis* and *Sarcocystis suihominis* can infect humans and both species utilize nonhuman primates as definitive hosts [82], dogs have no role yet in the zoonotic sarcocystosis.

**Table 10 animals-13-00496-t010:** Pooled prevalences estimated for protozoa detected in feces and/or intestinal scrapings of dogs from Egypt.

Parasite	No. Data Sets	No. Tested	No. Positive	Pooled Estimate % Based on 95% CI	Heterogeneity *I*^2^%
*Cystoisospora* spp.	10	5475	370	21.2 (−14.2–56.7)	99.97
*C. canis*	4	1178	163	23.3 (15.6–62.2)	99.9
*C. ohioensis*	3	998	119	12.2 (3.7–20.8)	95.88
*Sarcocystis* spp.	6	1126	29	2.0 (0.8–3.2)	36.84
*Cryptosporidium*	12	1754	111	5.5 (3.0–8.1)	89.25
*Giardia*	7	2699	210	7.4 (3.6–11.1)	94.33
*Entamoeba*	7	961	107	9.8 (4.1–15.5)	93.90

On the contrary, dogs have a potential role in human cryptosporidiosis, since *Cryptosporidium parvum* has been confirmed in dogs [83]. Oocysts of *C. parvum* were molecularly detected in feces of 2 out of 20 diarrheic puppies in Egypt [31], where cryptosporidiosis is common among humans [84]. In total, 12 datasets from Egypt tested 1754 dogs for *Cryptosporidium*; 111 were found infected, yielding a pooled prevalence of 5.5% (3.0–8.1%) (Table 10), which is slightly lower than that estimated for dogs worldwide (8.0%) [85]. Other protozoa that have potential health risks for humans were also detected in dogs from Egypt and showed estimated high pooled prevalences, e.g., *Giardia* spp. (7.4%, 3.6–11.1%) and *Entamoeba histolytica* (9.8%, 4.1–15.5%) (Table 10). Both parasites are prevalent among Egyptian children [86]. 

## 5. Conclusions

The present study provides an overview for various GIT parasites infecting dogs in Egypt based on various analyses for the published data addressing this topic. It is important to note that compared to reports from strays dogs in Dakahlia 40 years ago, the prevalence of these parasites has significantly decreased. However, the uneven geographical distribution and high heterogeneity of the published datasets represent possible limitations for this study. In addition, lack of sensitive diagnostic techniques (e.g., PCR) used to identify the parasite found to the species level is another limitation, for example, *T. multiceps*. While cerebral coenurosis is frequently detected in sheep from Egypt, eggs/adult worms of *T. multiceps* have not been identified in any survey on dogs from Egypt, leaving a gap in the epidemiology of this economically important parasite in this country. Despite these limitations, the present study was able to underline the role of dogs from Egypt in the epidemiology of various parasites with public health and veterinary importance.

## Figures and Tables

**Figure 1 animals-13-00496-f001:**
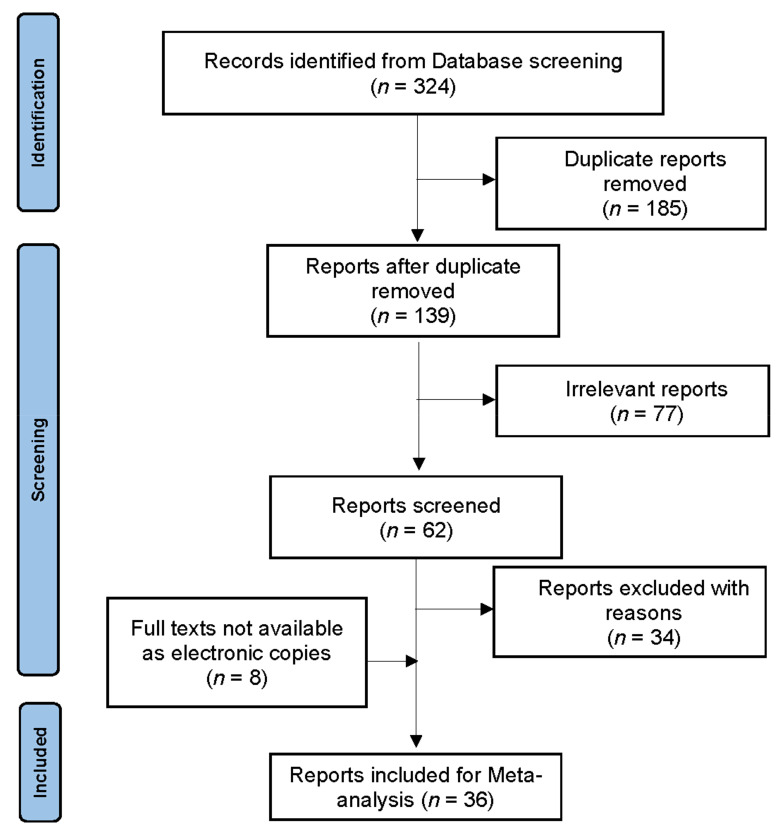
Flow diagram established according to PRISMA guidelines and showing methodologies of databases search and selection of eligible articles.

**Figure 2 animals-13-00496-f002:**
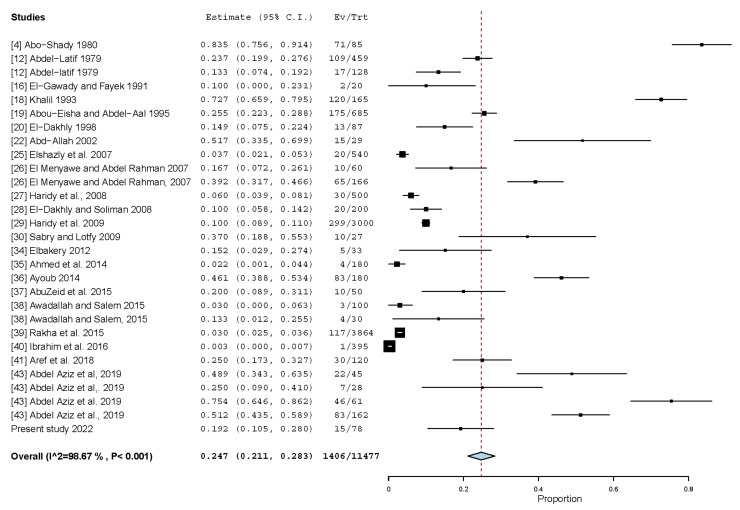
Forest plot diagram for random effects in the meta-analysis of the prevalence of *T. canis* in dogs from Egypt. The length of the line indicates 95% confidence interval of each study and the middle point of each line refers to the prevalence. Diamond refers to the overall prevalence.

**Figure 3 animals-13-00496-f003:**
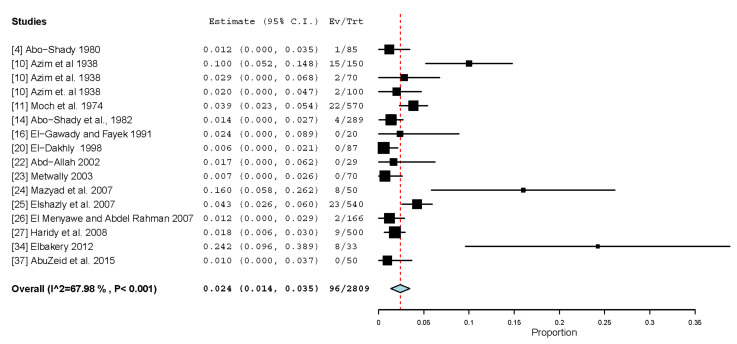
Forest plot diagram for random effects in the meta-analysis of the prevalence of *E. granulosus* in dogs from Egypt. The length of the line indicates 95% confidence interval of each study and the middle point of each line refers to the prevalence. Diamond refers to the overall prevalence.

**Figure 4 animals-13-00496-f004:**
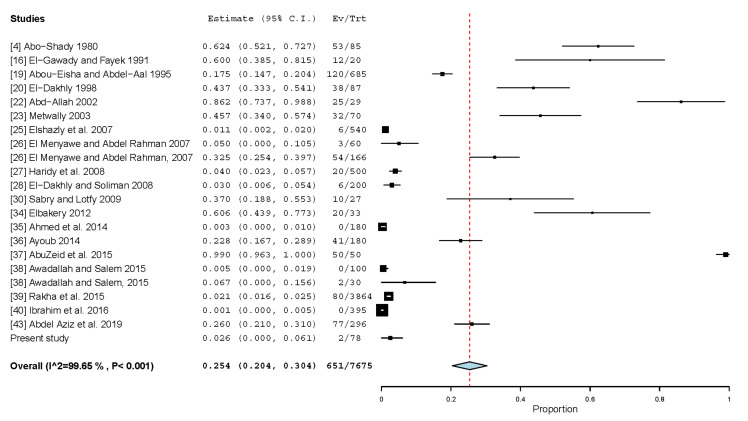
Forest plot diagram for random effects in the meta-analysis of the prevalence of *D. caninum* in dogs from Egypt. The length of the line indicates 95% confidence interval of each study and the middle point of each line refers to the prevalence. Diamond refers to the overall prevalence.

## Data Availability

On reasonable request, the corresponding authors will provide the datasets created and/or analyzed during the current work.

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
