# Peer review of "Gastrointestinal Parasites of Dogs in Egypt: An Update on the Prevalence in Dakahlia Governorate and a Meta-Analysis for the Published Data from the Country"

_animals, 2023, doi:10.3390/ani13030496_

Round 1
Reviewer 1 Report
The manuscript is interesting and relevant to veterinary sciences, especially in a country with a high disease burden like Egypt.
The reason for doing a meta-analysis for the study is not adequately understood. The study itself is pretty relevant. The authors must support their reasons.
The authors indicate, within their limitations, the use of different methods for the detection of parasites before the standard test, such as PCR, and nevertheless, in their study, they have used direct observation. Sustain the obstacles to doing the molecular tests.
I consider that this study is relevant to be included and discussed in your manuscript: https://doi.org/10.54034/mic.e1268
Figure 1 is pretty irrelevant.
Author Response
Response to the Comments from the editor and reviewers
Manuscript ID: animals-2073634
Type of manuscript: Article
Title: Gastrointestinal parasites of dogs: potential health risks for humans and animals in Egypt
Comments from the editors and reviewers:
First, we thank the editor and reviewers for providing constructive comments and suggestions, which were useful for improving the quality of the manuscript. We have now modified our manuscript in acknowledgement of all of the editor’s and reviewers’ comments and suggestions. The changes made are marked up using the “Track Changes” function. All references are relevant to the contents of the manuscript.
Reviewer-1
Comments and Suggestions for Authors
The manuscript is interesting and relevant to veterinary sciences, especially in a country with a high disease burden like Egypt.
Response:
Thanks so much for your motivating words
The reason for doing a meta-analysis for the study is not adequately understood. The study itself is pretty relevant. The authors must support their reasons.
Response:
This manuscript represents a hybrid of experimental research as well as a meta-analysis study. The latter is very important for evaluating the whole situation of a certain parasite in a specific area. Isolated surveys recruiting limited numbers of samples have been published to investigate GIT parasites infecting dogs in some Egyptian governorates. Many of these surveys have also been published in local journals not available for scientists worldwide. Therefore, combining results of these surveys with those of the present study in various meta-analyses would provide for the scientific community and health professionals a comprehensive overview on GIT parasites infecting dogs and their potential zoonoses in Egypt, clarified in the Introduction section
The authors indicate, within their limitations, the use of different methods for the detection of parasites before the standard test, such as PCR, and nevertheless, in their study, they have used direct observation. Sustain the obstacles to doing the molecular tests.
Response:
We were not, unfortunately, able to molecularly examine the collected samples for various GIT parasites, because of the very expensive molecular tests which are not available for many laboratories in the developing countries including Egypt.
I consider that this study is relevant to be included and discussed in your manuscript: https://doi.org/10.54034/mic.e1268
Response:
The study has been included in the meta-analyses and result for Giardia species has been modified, see Table 9.
Figure 1 is pretty irrelevant.
Response:
Figure 1 has been moved to the supplementary material (Fig. S1)

Reviewer 2 Report
Abbas et al. present an interesting analysis of the prevalence of zoonotic parasites from dogs in Egypt. The analysis of previously published information and information from the recent survey represent a valid approach. However, some aspects are ignored in the analysis , and also the presentation of the information is in need of considerable modification.
Starting with the analysis - the fact that information from as far back as 1938 was analyzed alongside studies from the last decade, puts doubt on the relevance of the pooled prevalence rates, which are the main reason for the metanalysis (since this is not a historical review). It can be assumed that some human, animal and environment related factors has changed to certain degrees along the time, as stated by the authors themselves when comparing recent and past information from the Dakahlia governorate. This point has not been addressed properly in the analysis and discussion. I suggest to consider analyzing information from the last 10-15 years, and comparing it to the analysis done to estimate if significant changes in trends is noticed. this would not only strengthen the results, but also be interesting on itself.
Regarding the presentation -
1. starting with the title, it reflects the general impression that authors have not decidec what this work is about - is it a small scale survey (which is in my eyes is much less interesting and important for an international audience), or a metanalysis of information concerning a whole country (which may be much more interesting, and a comparison point to other countries). I suggest to plavce the metanalysis at a pivotal point, and change the manuscript accordingly. As a start, a more informative title is advised, reflecting the fact that the work is mainly a metanalysis. In any case, current title is general and vague.
2. the abstract is altogether inappropriate and need to be re-written. Firstly, it does not mention the fact that a metanalysis was done ! and the transition between the results of the current small survey and metanalysis results is unclear. there is no mention of the time frame from which data was analyzed (studies between which years). there are no "bottom lines" (conclusions).
3. In the introduction , if echinococcosis is given as an example, provide some data on infection rates in humans, so readers can appreciate the scale of the potential zoonosis in humans. also, consider rewriting the paragraph on the objective(s) of this study, putting the metanalysis first,
3. the pictures of the parasite are redundant, overload the manuscript and can well be moved the supplementary material or removed altogether.
4. results of the metanalysis is given in the discussion instead of the results part.
5. forest plots are provided for some of the data only. the additional forest plots made should be made available in the supplementary material.
some comments on specific points
1. there is need for much editing and proofing of the manuscript. for eg., in many cases scientific names of parasites are either written in inverted commas (not needed) but not in italics (needed). in some places "prevalence" is used where "pooled prevalence" should be used. GRAM is gr. and not gm. , and more.
line 40 - the hyperlink does not lead to the relevant publication.
table 1 - for Toxocara, sum of cases of single or mixed infection is 14 where it should be 15.
line 71 - don't write "for eg." - state the exact data bases searched.
line 58 - do you have any information on the dogs that where sampled ? (estimated age, sex, approximate health status ?)
Table S1 - add year of publication for each reference.
Author Response
Response to the Comments from the editor and reviewers
Manuscript ID: animals-2073634
Type of manuscript: Article
Title: Gastrointestinal parasites of dogs: potential health risks for humans and animals in Egypt
Comments from the editors and reviewers:
First, we thank the editor and reviewers for providing constructive comments and suggestions, which were useful for improving the quality of the manuscript. We have now modified our manuscript in acknowledgement of all of the editor’s and reviewers’ comments and suggestions. The changes made are marked up using the “Track Changes” function. All references are relevant to the contents of the manuscript.
Reviewer 2
Comments and Suggestions for Authors
Abbas et al. present an interesting analysis of the prevalence of zoonotic parasites from dogs in Egypt. The analysis of previously published information and information from the recent survey represent a valid approach. However, some aspects are ignored in the analysis, and also the presentation of the information is in need of considerable modification.
Response:
Thank you so much for your motivating words
Starting with the analysis - the fact that information from as far back as 1938 was analyzed alongside studies from the last decade, puts doubt on the relevance of the pooled prevalence rates, which are the main reason for the metanalysis (since this is not a historical review). It can be assumed that some human, animal and environment related factors has changed to certain degrees along the time, as stated by the authors themselves when comparing recent and past information from the Dakahlia governorate. This point has not been addressed properly in the analysis and discussion. I suggest to consider analyzing information from the last 10-15 years, and comparing it to the analysis done to estimate if significant changes in trends is noticed. this would not only strengthen the results, but also be interesting on itself.
Thank you so much for this critical point. Prevalence variation overtime has been estimated for various helminth parasites tested in our analysis. Since the year of samples collection have not been provided in most reports from Egypt, datasets were classified into 2 groups according to the publishing year (before and after 2010), and various comparisons are performed, see tables 2 to 8; and sections 4.1 to 4.3.
Regarding the presentation -
- 1. starting with the title, it reflects the general impression that authors have not decided what this work is about - is it a small scale survey (which is in my eyes is much less interesting and important for an international audience), or a metanalysis of information concerning a whole country (which may be much more interesting, and a comparison point to other countries). I suggest to plavce the metanalysis at a pivotal point, and change the manuscript accordingly. As a start, a more informative title is advised, reflecting the fact that the work is mainly a metanalysis. In any case, current title is general and vague.
Response:
We fully agree with this comment. Therefore, title of the manuscript has been modified to be more informative for the work done. A new title “Gastrointestinal parasites of dogs in Egypt: An update on the prevalence in Dakahlia governorate and a meta-analysis for all published data from the country” is provided.
- 2. the abstract is altogether inappropriate and need to be re-written. Firstly, it does not mention the fact that a metanalysis was done! and the transition between the results of the current small survey and metanalysis results is unclear. there is no mention of the time frame from which data was analyzed (studies between which years). there are no "bottom lines" (conclusions).
Response:
The abstract has been modified.
- 3. In the introduction, if echinococcosis is given as an example, provide some data on infection rates in humans, so readers can appreciate the scale of the potential zoonosis in humans. also, consider rewriting the paragraph on the objective(s) of this study, putting the metanalysis first,
Response:
A few sentences have been provided in introduction section to describe cystic echinococcosis among humans and animals in Egypt. The objectives have been re-ordered to highlight the meta-analysis study.
- 3. the pictures of the parasite are redundant, overload the manuscript and can well be moved the supplementary material or removed altogether.
Response:
Figure 1 has been moved to the supplementary materials (Fig. S1).
- 4. results of the metanalysis is given in the discussion instead of the results part.
Response:
Results of the have been summarized in the discussion section to avoid repetition and longevity of the manuscript
- 5. forest plots are provided for some of the data only. the additional forest plots made should be made available in the supplementary material.
Response:
Done, see supplementary figures S2 – S5.
some comments on specific points
- 1. there is need for much editing and proofing of the manuscript. for eg., in many cases scientific names of parasites are either written in inverted commas (not needed) but not in italics (needed). in some places "prevalence" is used where "pooled prevalence" should be used.
Response:
Corrected
GRAM is gr. and not gm. , and more.
Response:
Corrected
line 40 - the hyperlink does not lead to the relevant publication.
Response:
A valid link has been provided.
table 1 - for Toxocara, sum of cases of single or mixed infection is 14 where it should be 15.
Response:
Corrected
line 71 - don't write "for eg." - state the exact data bases searched.
Response:
Corrected
line 58 - do you have any information on the dogs that where sampled? (estimated age, sex, approximate health status ?)
Response:
All dogs were in a good health status; however, data on the age and sex are of many dogs are missing.
Table S1 - add year of publication for each reference.
Response:
Provided in a separate column

Round 2
Reviewer 1 Report
I think that the responses have been satisfactory, although I still consider that the meta-analysis is not adequately supported. However, there is a situation that authors must consider: if they are going to continue with the meta-analysis in the main manuscript, ALL the studies that support this meta-analysis must be in the main manuscript and not in the supplementary files. The other option is to move all tables from the meta-analysis to the supplementary files and only mention them in the main manuscript.
Author Response
Response to the Comments from the reviewer
Manuscript ID: animals-2073634
Type of manuscript: Article
Title: Gastrointestinal parasites of dogs in Egypt: An update on the prevalence in Dakahlia governorate and a meta-analysis for the published data from the country
Comments from the reviewer:
First, we thank the reviewer for providing constructive comments and suggestions, which were useful for improving the quality of the manuscript. We have now modified our manuscript in acknowledgement of all of the editor’s and reviewers’ comments and suggestions.
Reviewer comment:
I think that the responses have been satisfactory, although I still consider that the meta-analysis is not adequately supported. However, there is a situation that authors must consider: if they are going to continue with the meta-analysis in the main manuscript, ALL the studies that support this meta-analysis must be in the main manuscript and not in the supplementary files. The other option is to move all tables from the meta-analysis to the supplementary files and only mention them in the main manuscript.
Response:
We have moved all studies of various meta-analysis conducted in the present manuscript from the supplementary data to the main manuscript, see table 10 in the revised manuscript.
